# Effect of Nylon Fiber Addition on the Performance of Recycled Aggregate Concrete

**Seungtae Lee**

Department of Civil Engineering, Kunsan National University, Kunsan 54150, Korea; stlee@kunsan.ac.kr;
Tel.: +82-10-3666-9561

**Abstract:** The adhered mortars in recycled aggregates (RA) may lower the performance of the concrete, by for instance reducing its strength and durability, and by cracking. In the present study, the effect of nylon fiber (NF) on the permeability as well as on the mechanical properties of concrete incorporating 100% RA was experimentally investigated. Concrete was produced by adding 0, 0.6 and 1.2 kg/m$^3$ of NF and then cured in water for a predetermined period. Measurements of compressive and split tensile strengths, ultrasonic pulse velocity and total charge passed through concrete were carried out, and the corresponding test results were compared to those of concrete incorporating crushed stone aggregate (CA). In addition, the microstructures of 28-day concretes were examined by using the FE-SEM technique. The test results indicated that recycled coarse aggregate concrete (RAC) showed a lower performance than crushed stone aggregate concrete (CAC) because of the adhered mortars in RA. However, it was obvious that the addition of NF in RAC mixes was much more effective in enhancing the performance of the concretes due to the crack bridging effect from NF. In particular, a high content of NF (1.2 kg/m$^3$) led to a beneficial effect on concrete properties compared to a low content of NF (0.6 kg/m$^3$) with respect to mechanical properties and permeability, especially for RAC mixes.

**Keywords:** recycled coarse aggregate concrete; nylon fiber; mechanical properties; permeability; microstructure

## 1. Introduction

Recently, a lack of natural aggregates with a high quality has set up the alarm to find alternative uses for recycled aggregate. Environmental and economic benefits led to a higher production and application of recycled aggregate concrete in many countries. A great amount of demolition and construction waste, almost 67,000,000 tons yearly, was generated in South Korea. For the purpose of the wide utilization of recycled aggregate, it is advised to use recycled aggregate up to 30% in concrete construction sites of 24 MPa grade and below [1]. Due to the increased technology in the aggregate production industry, both the quality and quality of recycled aggregates has increasingly improved [2].

However, there are still some doubts on the performance of concrete using recycled aggregate. In general, the performance of recycled aggregate concrete greatly depends on the adhered mortars oriented from parent concrete. Until now, there have been many reports on the properties of concrete made with recycled aggregate [3–11], but most of them resulted in a lower level of concrete strengths. This is mainly due to the residual impurities on the surface of the recycled aggregates, which blocked the strong bond between the cement paste and aggregate. Moreover, it is well known that there are two interfacial zones in recycled aggregate, which negatively affect concrete properties. In fact, this is the reason that the microstructure of recycled aggregate concrete is much more complicated than that of natural aggregate concrete. The old mortar in recycled aggregate includes many micro-cracks, formed during the production of recycled aggregate concrete, and it has a high porosity.

In order to enhance the performance of recycled aggregate concrete, researchers have come up with several advanced techniques. These techniques include the surface treatment [12], the use of mineral admixtures [13,14] and the addition of fibers [4,15,16], which can improve the performance of recycled aggregate concrete.

In particular, the reinforcement of recycled aggregate concrete using fibers leads to reduced micro-cracks in the cement matrix as well as to an increase in material density. There have been numerous studies on the applications of steel fiber [17–23] and polypropylene (PP) fibers [24–26] in concrete. Moreover, the PP fiber also enjoys popularity in the domain of recycled aggregate concrete [4,27]. However, although nylon fiber (NF) shows a rising acceptance in the literatures [28–30], it remains unpopular compared to steel fiber and PP fiber. In comparison with steel and PP fibers, while there is limited literature available on the use of NF, some authors [28,31] reported that the use of NF stepped up the performance after the presence of cracks in concrete, and sustained high stresses. One needs to examine the applicability of NF for the purpose of the enhanced mechanical properties and the reduced micro-cracks in recycle aggregate concrete.

This study is therefore aimed at investigating the usability of NF in recycled aggregate concrete in order to be used in structural concrete, since the use of NF in field concrete is gaining popularity nowadays. In order to achieve this goal, measurements of compressive strength, split tensile strength, ultrasonic pulse velocity, and chloride ion permeability of recycled coarse aggregate concretes with or without NF were carried out, and the corresponding test results were compared to those of concretes incorporating crushed stone aggregate. Additionally, the microstructures of 28-day concretes were examined by using the FE-SEM technique.

## 2. Experimental Section

### 2.1. Materials

In this study, ordinary Portland cement conforming to ASTM C150 was used in preparing the concrete specimens. The cement had been produced by a local cement plant in South Korea. The density and specific surface area of the cement used were 3.15 g/cm$^3$ and 328 m$^2$/kg, respectively. The mineralogical compounds of the cement were 54.9, 16.6, 10.3, and 9.1% for $C_3S$, $C_2S$, $C_3A$ and $C_4AF$, respectively.

Crushed stone aggregate (CA) and recycled coarse aggregate (RA) were used as coarse aggregates for the concrete production. RA was produced by crushing the waste concrete with a jaw crusher and an impact crusher. To enhance the purity of RA, impurities such as wood, bricks and glass were manually removed. A water jet with high-pressure was also used to remove mud and debris from RA. Both CA and RA have continuous grains from 5 to 25 mm. The main properties of coarse aggregates are presented in Table 1. The size grading of CA and RA is shown in Figure 1. Natural river sand was used as a fine aggregate with a fineness modulus of 2.80, a water absorption of 0.98% and a density of 2.65 g/cm$^3$.

**Table 1.** Physical properties of the coarse aggregates.

| Properties | Crushed Stone Aggregate (CA) | Recycled Coarse Aggregate (RA) |
|---|---|---|
| Density (g/cm$^3$) | 2.64 | 2.37 |
| F.M. | 7.17 | 7.43 |
| Absorption (%) | 0.66 | 4.31 |
| Abrasion rate (%) | 21.7 | 47.5 |
| Adhered mortar (%) [1] | | 5.56 |

[1] Acid-soluble content.

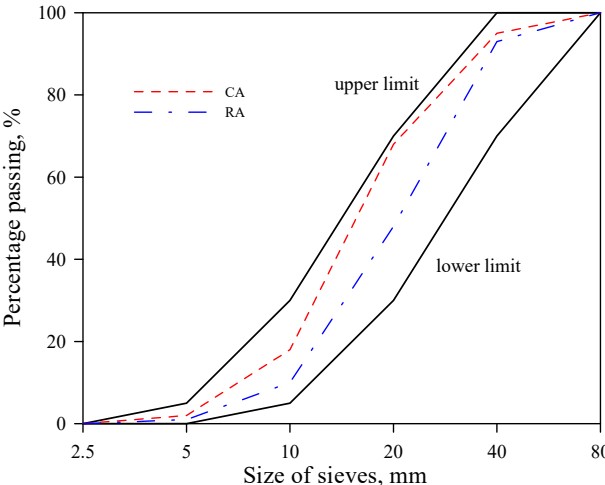

**Figure 1.** Size grading of coarse aggregates.

The nylon fibers (NF) used in the present study, a picture of which is shown in Figure 2, had been supplied by a local fiber company in South Korea, and they are now commercially available in the domestic market. The properties of NF used in the present study are shown in Table 2. In addition, a polycarboxylate-based superplasticizer (SP) was used to improve the initially low workability of the fresh concrete. The basic properties of SP were provided by a chemical manufacturer in South Korea.

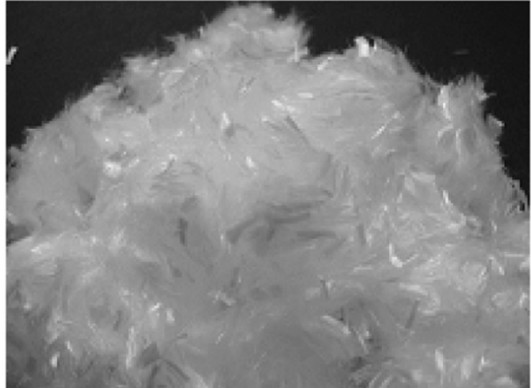

**Figure 2.** Nylon fibers (NF) used in this study.

**Table 2.** Physical properties of NF.

| Properties | Nylon Fibers (NF) |
|---|---|
| Diameter ($\mu$m) | 23 |
| Length (mm) | 19 |
| Aspect ratio | 826 |
| Density (g/cm$^3$) | 1.16 |
| Tensile strength (MPa) | 919 |
| Elastic modulus (GPa) | 5.3 |
| Color | white |

*2.2. Concrete Mix Proportions and Curing*

Six concrete mixes of 24 MPa grade concrete have been prepared. For all specimens, the w/c ratio was chosen as 0.50. The desired workability which is the range of 130–170 mm collapse was obtained by means of SP. The RAC was prepared with a 100% replacement of RA with CA. The concrete mix proportions are shown in Table 3. The NF was added at the concentrations of 0, 0.6, and 1.2 kg/m$^3$ for CAC and RAC mixes, respectively. The SP was initially mixed with water to achieve a uniform

dispersion. The concrete mixes were grouped to study the effect of the NF contents. The mixing of concrete was carried out using a manually loaded laboratory mixer. The concrete materials were mixed and then cast into the $100 \times 200$ mm cylinder molds for compressive and split tensile strength, non-destructive ultrasonic pulse velocity measurements, and a rapid chloride ion penetration test. The concrete specimens were cured under moisture conditions for 24 h, after which they were demolded and moved into a plastic tank for water curing until the time of testing.

**Table 3.** Mix proportions of concrete.

| Concrete Mixes | Mix Proportion (kg/m$^3$) | | | | | | | Fresh Density [1] (kg/m$^3$) |
|---|---|---|---|---|---|---|---|---|
| | Water | Cement | Sand | CA | RA | NF | SP [3] | |
| CAC1 | 170 | 340 | 745 | 1015 | - | - | 2.45 | 2345 |
| CAC2 | 170 | 340 | 745 | 1015 | - | 0.6 (0.06) [2] | 2.45 | 2350 |
| CAC3 | 170 | 340 | 745 | 1015 | - | 1.2 (0.12) | 2.45 | 2386 |
| RAC1 | 170 | 340 | 745 | - | 912 | - | 2.72 | 2278 |
| RAC2 | 170 | 340 | 745 | - | 912 | 0.6 (0.06) | 2.72 | 2290 |
| RAC3 | 170 | 340 | 745 | - | 912 | 1.2 (1.12) | 2.72 | 2312 |

[1] Values measured according to ASTM C 138M-17a standard [32]. [2] Fiber volume fraction (v/v %). [3] Superplasticizer.

### 2.3. Test Methods

The compressive and split tensile strengths of concrete cylinders with dimension of 100 mm in diameter and 200 mm in length after 7, 28, and 91 days of curing were tested according to ASTM C39 [33] and ASTM C469-17 [34], respectively. For evaluating the compressive strength, the cylinders were placed under a compression testing machine of 2000 kN capacity. The test for split tensile strength was also conducted using the same compression testing machine. The mean values of the compressive and split tensile strengths of at least three samples for each concrete mix were taken, and the standard deviation from the test results was calculated. The non-destructive test, like the ultrasonic pulse velocity (UPV), was conducted according to ASTM C597-16 [35] by using PUNDIT LAB, manufactured by PCTE, Australia. The used transducers are 50 mm in diameter, and have a maximum resonant frequency of 54 kHz. A rapid chloride ion penetrability test was used to evaluate the permeability of the concrete specimens. This test was based on the standard test method of ASTM C1202-18 [36]. After having been cured for 28 days, a concrete disc, 50 mm in thickness and 100 mm in diameter, was connected to two chambers: one was filled with 3% NaCl solution and the other with 0.3M NaOH solution to form electrodes, as shown in Figure 3. An electric charge of 60 V was applied to the electrodes for 6 h. The current flowing through the concrete disc and the temperature of the solution in the chambers were recorded with an interval of 30 min. The higher level of the total charge passed represents the higher permeability of concrete, as shown in Table 4.

The microstructures of concrete fractions after the compressive strength tests of 28-day concrete specimens, were investigated using field emission scanning electron microscopy (FE-SEM) equipped with an EDXA Falcon energy system. The concrete samples were gold-coated after drying in a desiccator for 24 h.

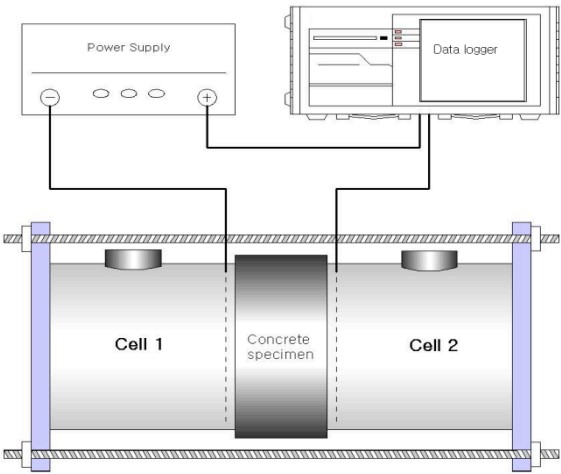

**Figure 3.** Schematic of the rapid chloride ion penetration test based on ASTM C1202-18 [36].

**Table 4.** Chloride ion penetrability based on the charge passed [36].

| Charge Passed (Coulombs) | Chloride Ion Penetrability |
|:---:|:---:|
| >4000 | High |
| 2000–4000 | Moderate |
| 1000–2000 | Low |
| 100–1000 | Very low |
| <100 | Negligible |

## 3. Results and Discussion

### 3.1. Compressive Stregnth

Table 5 presents the compressive strength values of both CAC and RAC mixes with different NF content. Due to the adhered mortars in recycled aggregates, the compressive strength values of RAC mixes is much lower than CAC mixes. For example, it can be seen that the compressive strength value of the RAC1 mix without NF decreased by 27% at the age of 28 days, compared with that of the CAC1 mix. At 91 days, the CAC1 mix also exhibited a higher compressive strength value, showing 41.2 MPa, than the RAC1 mix (30.1 MPa). It is presumed that the higher absorption and the larger pores, due to the adhered mortar in the RAC1 mix, contributed to a decreased strength in the RAC1 mix.

The compressive strength generally increased when NF was added to the mixes. Among the concrete mixes with RA, the highest compressive strength was obtained from the RAC3 mix with 52.6 MPa at 91 days of curing. For the CAC mixes, the CAC3 mixes exhibited a good development in compressive strength. It was therefore obvious that the addition of NF led to the increase of compressive strength in both the CAC and RAC mixes. It is worth noting that despite the use of RA, the RAC3 mix with 1.2 kg/m$^3$ of NF showed much higher compressive strength values, ranging from 27~41%, compared with the CAC1 mix with CA, for all stages of the curing.

In order to highlight the effect of RA and NF on the compressive strength of concrete mixtures, the compressive strength ratio (CSR) of concrete mixes at 7, 28 and 91 days was calculated, and the correspondent results were shown in Figure 4a–c, respectively. The CSR results shown in Figure 4 exhibited almost the same trend, regardless of the curing ages. However, it must be noted that when NF was added in the RAC mixes, there was a beneficial effect on the increase in compressive strength, as a similar result was reported by Song et al. [37]. They found an approximately 11.5% increase in the compressive strength of concrete with 0.6 kg/m$^3$ of NF content, compared to control concrete without NF.

From Figure 4, it was clearly observed that the CSR increased when the NF contents increased. Increases of CSR in the high content NF containing mixes (CAC3 and RAC3) were more remarkable

than for the low content NF mixes such as CAC2 and RAC2. At all curing periods, the higher CSR values were obtained from the RAC3 mix with an NF content of 1.2 kg/m$^3$, which means that, as the NF content increases, the compressive strength development of recycled aggregate concrete becomes more effective. For example, the CSR of the RAC3 concrete was 194% at 7 days, while the value for the CAC3 mix was only 136%. More importantly, the improved behavior of concrete with NF may be the result of the high content of fibers which form a network that acts as a bridge in the cement matrix, resulting in a reduction of micro-cracks [16,27,38].

**Table 5.** Compressive strength values of concrete mixes with different NF contents (standard deviation in parenthesis).

| Mixes | Compressive Strength (MPa) | | |
|---|---|---|---|
| | 7d. | 28d. | 91d. |
| CAC1 | 25.6 (0.97) | 36.5 (1.14) | 41.2 (1.12) |
| CAC2 | 37.5 (1.42) | 45.6 (1.02) | 52.4 (0.76) |
| CAC3 | 38.2 (0.88) | 49.8 (1.65) | 56.6 (1.35) |
| RAC1 | 18.6 (1.21) | 25.4 (2.18) | 30.1 (0.90) |
| RAC2 | 27.0 (0.45) | 39.5 (0.98) | 43.6 (2.33) |
| RAC3 | 36.2 (0.81) | 47.2 (1.47) | 52.6 (0.66) |

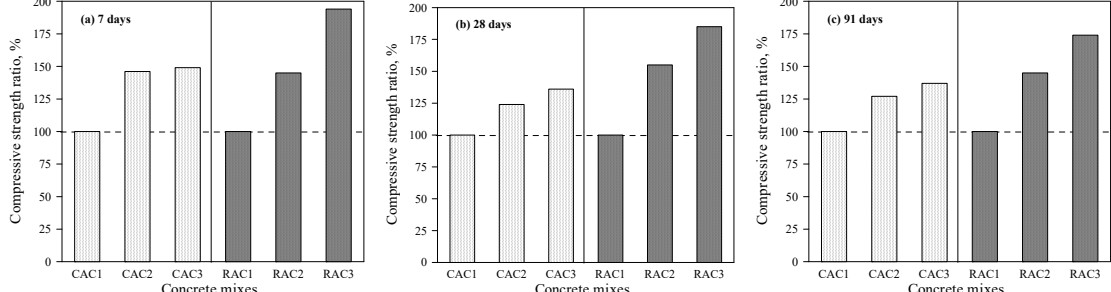

**Figure 4.** Compressive strength ratios of concrete mixes at: (**a**) 7 days, (**b**) 28 days, and (**c**) 91 days.

## 3.2. Split Tensile Strenth

The split tensile strength values of the CAC and RAC mixes with different NF contents at 7, 28, and 91 days are listed in Table 6. As expected, the values of the split tensile strength of concrete were significantly dependent on the added NF content. The strength results showed that there was a great increase in the split tensile strength values with the addition of NF up to 1.2 kg/m$^3$ in both the CAC and RAC mixes. At 91 days, the maximum strength value obtained in the case of the CAC mixes was 6.3 MPa for 1.2 kg/m$^3$ NF content, an increase of 23.5% over the CAC1 mix. Similarly, for the RAC mixes, the maximum improvement examined in the RAC3 mix with the same NF content, as compared with the unreinforced RAC1 mix, was 80.6%, which has a split tensile strength of 5.6 MPa. These results were in a good agreement with other studies [15,27].

The calculated results of the split tensile strength ratio (SSR) of the concrete mixes at 7, 28 and 91 days were presented in Figure 5a–c, respectively. For both the CAC and RAC mixes, it seems that the addition of NF had much influence on the split tensile strength, especially at the early age of curing. At 7 days, the SSR value for the RAC3 mix was 205% compared to 151% for the CAC3 mix, and similar trends were also observed at 28 and 91 days, as shown in Figure 5b,c.

Within the scope of the present study, it was confirmed that a higher addition of NF led to an increase in both the compressive and split tensile strengths. Furthermore, this was obvious when NF was applied in the RAC mixes compared to the CAC mixes.

**Table 6.** Split tensile strength values of the concrete mixes with different NF contents (standard deviation in parenthesis).

| Mixes | Split Tensile Strength (MPa) | | |
|---|---|---|---|
| | 7d. | 28d. | 91d. |
| CAC1 | 2.9 (0.14) | 4.2 (0.10) | 5.1 (0.22) |
| CAC2 | 4.2 (0.20) | 5.2 (0.24) | 6.0 (0.24) |
| CAC3 | 4.4 (0.42) | 5.6 (0.32) | 6.3 (0.33) |
| RAC1 | 1.8 (0.12) | 2.5 (0.14) | 3.1 (0.24) |
| RAC2 | 3.1 (0.22) | 3.9 (0.11) | 4.7 (0.10) |
| RAC3 | 3.7 (0.18) | 4.7 (0.23) | 5.6 (0.19) |

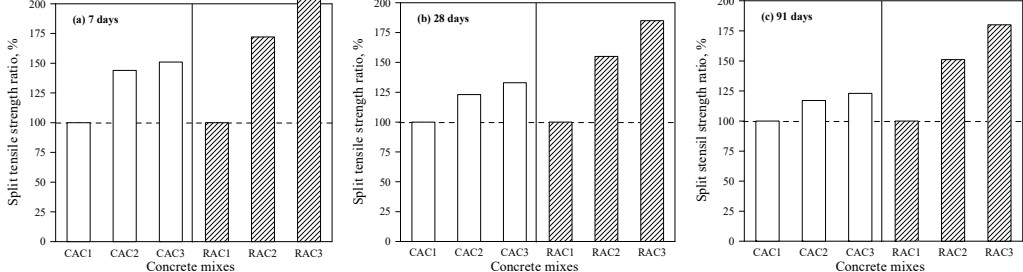

**Figure 5.** Split tensile strength ratio of concrete mixes at: (**a**) 7 days, (**b**) 28 days, and (**c**) 91 days.

### 3.3. Ultrasonic Pulse Velocity

The ultrasonic pulse velocity (UPV) measurements have been well known as non-destructive methods to assess the quality of concrete [14,27,39,40]. The UPV values for the CAC and RAC mixes measured at 7, 28, and 91 days are shown in Figure 6. According to the classification criterion for concrete based on ultrasonic pulse measurements by Leslie and Cheeseman [41], the UPV values for concrete mixes observed in this work can be classified as shown in Table 7.

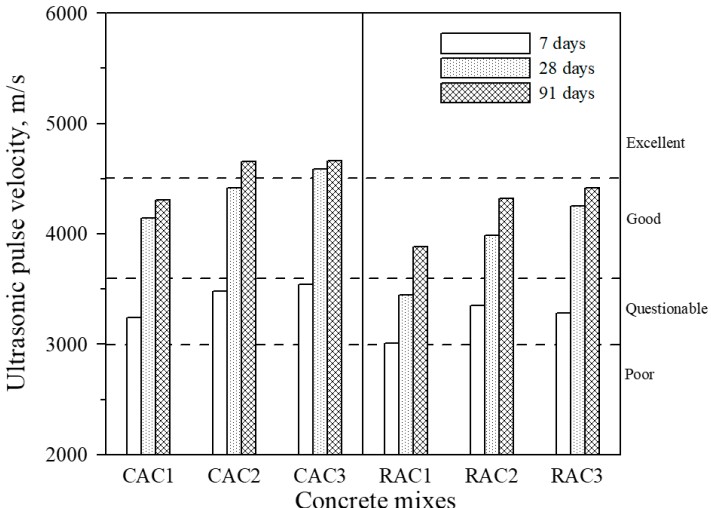

**Figure 6.** Ultrasonic pulse velocity of concrete mixes.

**Table 7.** Concrete classification based on the ultrasonic pulse velocity [41].

| Pulse Velocity (m/s) | Concrete Classification |
|---|---|
| V > 4500 | Excellent |
| 3600 < V < 4500 | Good |
| 3000 < V < 3600 | Questionable |
| 2100 < V < 3000 | Poor |
| V < 2100 | Very poor |

As expected, it was observed from Figure 6 that the UPV values increased with curing time. The increase of UPV with time is a natural development due to the increase of the stiffness via the hydration reaction [42]. Among the concrete mixes made with CA (CAC mixes), the UPV values for the CAC2 mix were almost similar to those for the CAC3 mix, which shows that the content of NF in the CAC mixes does not have a significant influence in the variation of the UPV. However, the CAC2 and CAC3 mixes exhibited a better performance with respect to the UPV compared to the CAC1 mix without NF. This may be partially attributed to the bridge effect of NF which leads to the reduction of micro-cracks in the cement matrix by the addition of NF [29,30,43]. Generally, with an addition of steel fiber, introducing steel fibers of greater length negatively affected the UPV of the concrete specimens [44]. However, in this study, it can be seen that introducing NF positively affects the UPV, although it was relatively long (19 mm in length). This might be attributed to the increase of the materials density and the bridge effect of the fibers, due to incorporation of NF into the concrete mixes. In the case of the RAC mixes, with the further incorporation of NF, the general increases in the UPV were also examined, showing similar trends with regards to the strength properties (Tables 5 and 6) with the increase of NF. At 7 days, the UPV values of the RAC mixes were comparable with the CAC mixes, ranging from 3240 ~ 3540 m/s. This can be attributed to the substantially higher water absorption capacity of RA (4.31%) than CA (0.66%), because of the adhered mortar in RA, resulting in the increased porosity in the RAC mixes. However, relatively small increases in the UPV were observed in the RAC mixes at later ages of curing. Similar to the results of the compressive and split tensile strengths, the UPV results indicated that the CAC mixes showed higher values in the UPV than the RAC mixes, regardless of the aggregate types and NF contents.

The relationship between compressive strength (CS) and split tensile strength (SS), and the UPV obtained from the present study, are presented in Figure 7. It can be seen from this figure that the best fit-curve representing the relationship is given as;

UPV = 2450 Exp (0.012 × CS), determined by a proposed regression model of $R^2$ = 0.82.

UPV = 2526 Exp (0.1016 × SS), determined by a proposed regression model of $R^2$ = 0.84.

High correlation coefficient values (0.82 for UPV-CS curve, and 0.84 for UPV-SS curve) were obtained from the exponential curves, implying that the trend of the UPV values is almost similar to that of both the compressive strength and split tensile strengths [44,45]. Overall, it can be said that the non-destructive UPV measurements are a useful method to determine the mechanical properties of concrete.

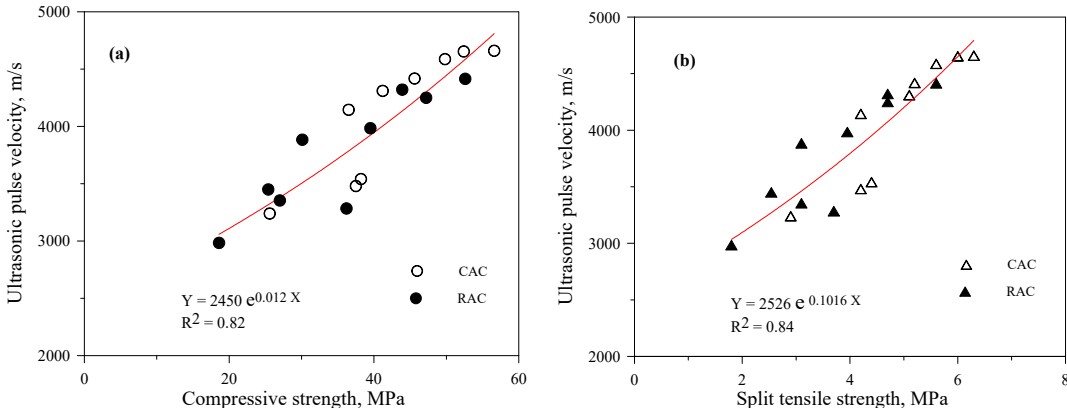

**Figure 7.** Relationship between the strengths and ultrasonic pulse velocity of concrete mixes: (**a**) compressive strength, and (**b**) split tensile strength.

### 3.4. Rapid Chloride Penetration Test

Based on the ASTM C1202-18 standard [36], the results of the rapid chloride ion penetration test (RCPT) were presented in Figure 8, indicating the total charge that passed through the concrete samples. It can be seen that NF in the CAC mixes reduced the total charge, compared to the CAC1

mix without the addition of NF, and that the reduction of the total charge was more remarkable in the concrete mixes with RA. It was observed that the charge passed for the 28-day RAC mixes were 2872, 2294, and 2050 coulombs for the RAC1, RAC2, and RAC3 mixes, respectively. According to the ASTM C1202-18 criterion, the total charges for all the RAC mixes corresponded to the level of 'Moderate', while they were 'Low' for the CAC mixes, ranging from 1046 to 1424 coulombs (see Table 4). This implies that RA in concrete may impose a high risk to the durability of the concrete structure with respect to steel corrosion. However, the usage of NF in concrete made with RA can mitigate the possibility of steel corrosion oriented from external chlorides due to the reduction of micro-cracks in the cement matrix.

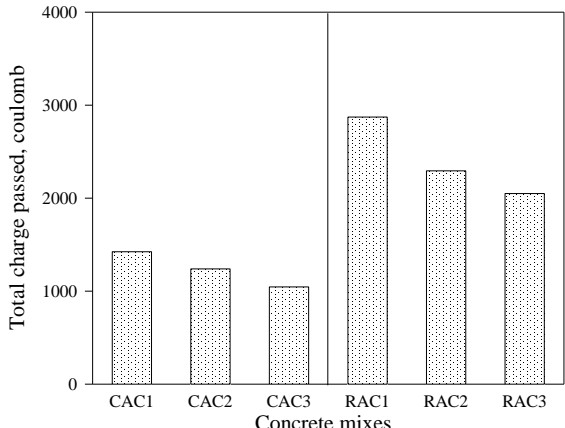

**Figure 8.** Rapid chloride ion penetration test (RCPT) results of the concrete mixes.

*3.5. Microstructure*

The results of the microstructural observations of the concrete samples using the FE-SEM technique are shown in Figures 9–14. The analysis was performed on fractions of the CAC and RAC specimens obtained after the compressive strength of concretes, to observe the effect of NF addition. It is well understood that the interfacial transition zone (ITZ) affects the mechanical properties and durability of the concrete [46–49]. In the case of the CAC1 sample (Figure 9), it was found from the image that there were calcium hydroxide (CH) crystals and C-S-H gel, in addition to a small amount of ettringite (AFt) and monosulfate (AFm). Furthermore, the ITZ between the aggregate and bulk cement matrix seems to be dense, indicating a lower porosity and less cracks. The strengthening of the ITZ connection results in the microstructural integrity of the cement matrix. The images for the CAC samples with NF (CAC2 and CAC3) were also examined via the FE-SEM technique, and revealed that C-S-H gel with a dense structure was mainly present through the samples, as shown in Figures 10 and 11.

Comparatively, for the microstructure of the RAC1 samples (Figure 12), there are two interfacial zones; the old ITZ between the aggregate and adhered mortar, and the new ITZ between t recycled aggregate and new mortar. The adhered mortar can be easily differentiated from the new mortar by different degrees of hydration of the cement matrix [27]. It was obvious from the microstructural observation that the crack was formed along this weak interface. This results in both a lower strength and higher permeability in the RAC sample, as shown in Tables 5 and 6, and in Figure 8.

The two interfacial zones were also observed in the microstructure of the RAC2 sample, as shown in Figure 13. Due to the addition of NF, the sample exhibited the cement bulk matrix with a higher density as well as with less pores. With the NF addition of 1.2 kg/m$^3$ in the RAC mix, as shown in Figure 14, only small cracks were found on the paste of the specimen, which is due to the bridge effect of NF. In addition, the increase in density (see Table 3) of the RAC mixes with NF (RAC2 and RAC3) led to the improvement of mechanical properties such as the strength and UPV as well as to the

reduction of permeability. Therefore, it can be confirmed that the enhanced performance of the RAC3 mix compared to the RAC1 mix is due to the reduced micro-cracks via the increase in the fiber content.

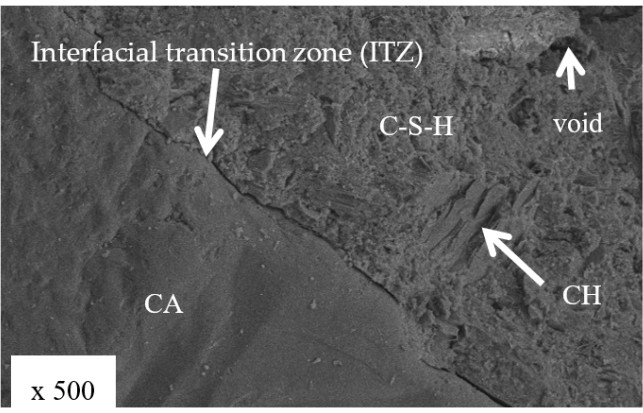

**Figure 9.** FE-SEM image of CAC1 sample.

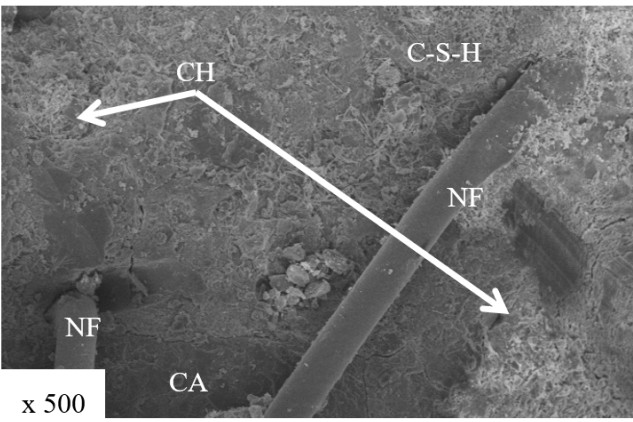

**Figure 10.** FE-SEM image of CAC2 sample.

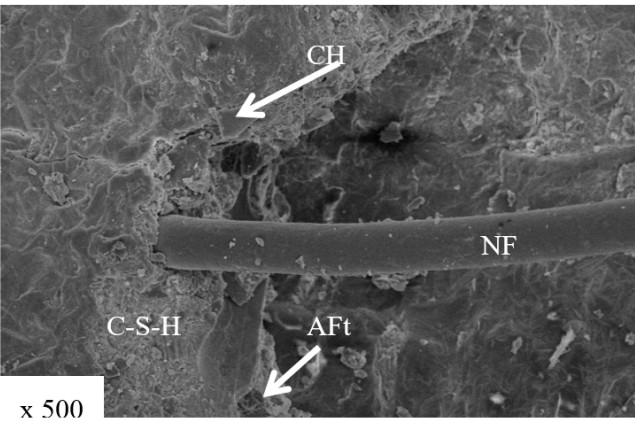

**Figure 11.** FE-SEM image of CAC3 sample.

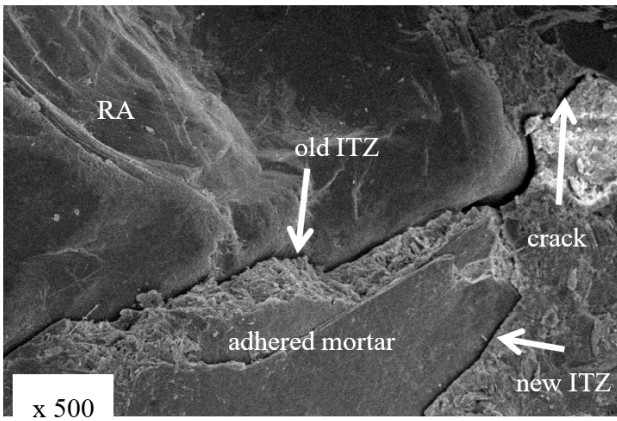

**Figure 12.** FE-SEM image of RAC1 sample.

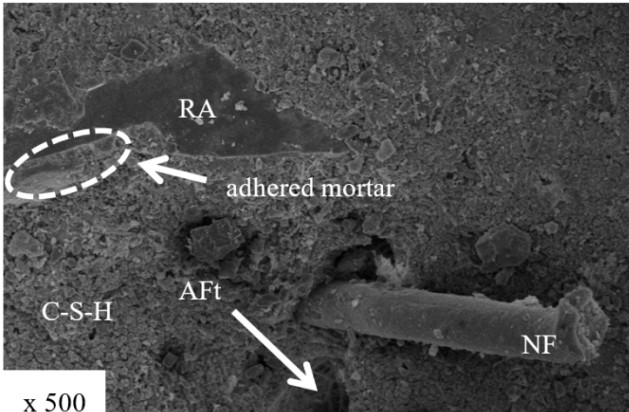

**Figure 13.** FE-SEM image of RAC2 sample.

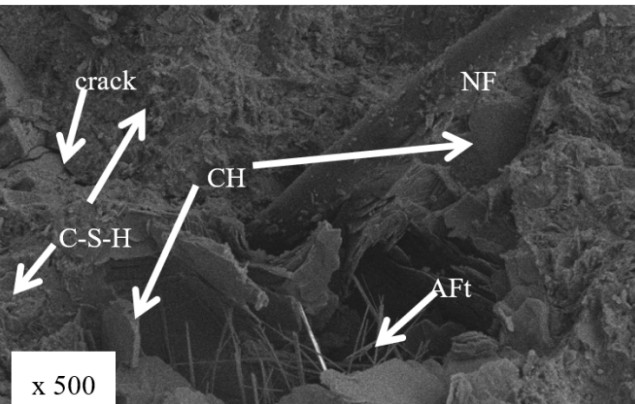

**Figure 14.** FE-SEM image of RAC3 sample.

## 4. Conclusions

Experimental works were carried out with the additions of 0, 0.6 and 1.2 kg/m$^3$ of NF in both CAC and RAC mixes in order to highlight the effect of NF on the mechanical properties of CAC and RAC mixes. The RAC was prepared with a 100% replacement of RA with CA. The main conclusions obtained from the present study are as follows.

(1)　Due to the adhered mortar in RA, the compressive strength values of the RAC mixes were significantly lower than those of the CAC mixes. However, we found that the addition of NF led to an increase in compressive strength of both the CAC and RAC mixes. In particular, this trend

was more remarkable in the RAC mixes with a high content of NF. The compressive strength ratio results revealed that there was a beneficial effect of NF on the increase in compressive strength.

(2)     As observed in the case of the compressive strength, a similar trend was also examined with the NF content variation for the split tensile strength. Specifically, the test results revealed that there was a significant increase in the split tensile strength, especially with the addition of 1.2 kg/m$^3$ NF, regardless of concrete types. In the case of the RAC3 mix, we examined an increase of 80.6% in the split tensile strength over the RAC1 mix without NF.

(3)     From the test results of the UPV, we observed that in the case of the RAC mixes, there was an increase in the UPV with the further incorporation of NF. This may be due to the increase of the materials density and the bridge effect of NF. Additionally, it seems that the UPV results were closely related to those of both the compressive and split tensile strengths observed, with high correlation coefficient values.

(4)     Based on the results of RCPT, we observed that the addition of NF in the CAC mixes reduced the total charge, compared to the control (CAC1) mix without the addition of NF, and the reduction of the total charge was more remarkable in the concrete mixes with RA. This implies that the usage of NF in the RAC mixes can mitigate the possibility of steel corrosion oriented from external chlorides due to the reduction of micro-cracks in the cement matrix.

(5)     The microstructural observation of concrete revealed that the micro-cracks propagated along the ITZ between old mortar and aggregate, especially in the RAC mixes. However, for RAC mix with an addition of NF, the NF plays an important role in crack bridging, resulting in a higher strength and lower permeability in concrete.

Within the scope of this study, we can conclude that the addition of NF enhanced the permeability as well as the mechanical properties, especially in concrete incorporating RA. The enhancement is primarily attributed to the bridge effect of NF, which allowed for a higher development of strength and concrete density.

**Funding:** This research was funded by the Ministry of Land, Infrastructure and Transport (MOLIT) and the Korea Agency for Infrastructure Technology Advancement (KAIA) for a project on "Development of manufacture technology for high-quality recycled aggregate and application of concrete (No. 16C16020280)".

**Conflicts of Interest:** The author declares no conflicts of interest.

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
