# Peer review of "Effect of Nylon Fiber Addition on the Performance of Recycled Aggregate Concrete"

_applsci, doi:10.3390/app9040767_

Round 1

Reviewer 1 Report

The paper deals with the study of the mechanical properties of concrete containing recycled aggregate reinforced with nylon fiber. The study is well done and the goal of the research seems reached. In the section 2.3 the authors should specify how many specimens have been realized, moreover some images of the specimens after the tests would be appreciated. In table it is not clear if the compressive strength represent a mean or a characteristic value. In both the cases, the CoV should be represented. The same goes for tensile strength and the table. It would be interesting to see the FE-SEM image of all the mixture (not only CAC1, RAC1, RAC3). Finally, to investigate the durability, also rapid carbonation test would be useful.

Author Response

Please find attachement.

Reviewer 2 Report

The paper studies the effect of nylon fiber addition on the performance of recycled aggregate concrete. The paper is reasonably well organized, written and illustrated, and provides a clear discussion. The reviewer thinks that the paper can be published, even though the authors should address the aspects below before publication:     

1)  Introduction: The sentence “There have been numerous studies on the applications of steel fiber in concrete [17,18]” should be replaced by: “There have been numerous studies on the applications of steel fiber [] and PP fibers in concrete []”.

This sentence should be supported by new citations on the use of both fibers.

About steel fibers please consider the following research works:

Shear Behaviour of Steel Fibre-Reinforced Concrete Beams without Stirrup Reinforcement”, ACI Structural Journal

Shear behavior of prestressed precast beams made of self-compacting Fiber Reinforced Concrete”, Construction and Building Materials

Self-healing capacity of fiber reinforced cementitious composites. State of the art and perspectives

A methodology to assess crack-sealing effectiveness of crystalline admixtures under repeated cracking-healing cycles. Construction and Building Materials

Structural behaviour of precast tunnel segments in fiber reinforced concrete. Tunnelling and Underground Space Technology

About PP fibers please consider the following research works:

An experimental study on the shear behaviour of reinforced concrete beams with macro-synthetic fibres.  Construction and Building Materials

Shear behaviour of prestressed double tees in self-compacting polypropylene fibre reinforced concrete. Engineering Structures

Influence of steel and macro-synthetic fibers on concrete properties. Fibers

Plastic fibres as the only reinforcement for flat suspended slabs: experimental investigation and numerical simulation. Construction and Building Materials

2) Line 84 and line 89: avoid in the manuscript to use commercial names of companies.

3) Table 2: add fiber aspect ratio.

4) Line 105-106: Please better discuss why 24 MPa is a grade concrete typical for structural applications.

5) Table 3: add fiber volume fraction (%)

6) Did the Authors carry out any mechanical characterization of fiber reinforced concretes? For example according to ASTM1609 or EN 16451.

Round 2

Reviewer 1 Report

The reviewer thanks the authors for the effort made in the correction of the paper